# Simulation Method for the Impact of Atmospheric Wind Speed on Optical Signals in Satellite–Ground Laser Communication Links

**Wujisiguleng Zhao [1,2] and Chunyi Chen [1,\*]**

1   School of Computer Science and Technology, Changchun University of Science and Technology, Changchun 130022, China; jsfz@hlbec.edu.cn
2   School of Artificial Intelligence and Big Data, Hulunbuir University, Hulunbuir 021008, China
\*   Correspondence: zsb@cust.edu.cn

**Abstract:** To analyze the intensity of atmospheric turbulence in a satellite–ground laser communication link, it is important to consider the effect of increased atmospheric turbulence caused by wind speed. Atmospheric turbulence causes a change in the refractive index, which negatively impacts the quality and focusing ability of the laser beam by altering its phase front. To simulate the changes in amplitude and phase characteristics of laser beam propagation in atmospheric turbulence caused by wind speed, a transverse translation phase screen is used. To better understand and address the influence of atmospheric wind speed on the phase of optical signals in satellite–ground laser communication links, this paper proposes a Monte Carlo simulation method. This method utilizes the spatial and temporal variations in the refractive index in the atmosphere and integrates the principles of optical signal propagation in the atmosphere to simulate changes in the phase of optical signals under different wind speed conditions. By analyzing the variations in the received optical signal's power, the Monte Carlo method is employed to simulate phase screens and logarithmic amplitude screens. Additionally, it models the probability density of the statistical behavior of received optical signal's fluctuations, as well as the time autocorrelation coefficient of optical signals. This paper, under the coupling condition in satellite–ground laser communication links, conducted a Monte Carlo simulation experiment to analyze the characteristics of the optical signal's fluctuations in the link and discovered that atmospheric wind speed affects the shape of the power spectral density model of the received optical signal. Increasing wind speed leads to a decrease in the time autocorrelation coefficient of the received optical signal and affects the coupling efficiency. The paper then used a cubic spline interpolation fitting method to verify the models of the power spectral density and the autocorrelation time coefficient of the optical signal. This provides a theoretical foundation and practical guidance for the optimization of satellite–ground laser communication systems.

**Keywords:** atmospheric wind speed; satellite–ground laser communication link; simulation; phase screen

## 1. Introduction

With the swift evolution of modern communication technologies, satellite–ground laser communication has increasingly become a crucial technology for meeting the demands of high-capacity data transmission. Possessing both high bandwidth and low latency, it offers an efficient means for communicating vast amounts of information with minimal delay. In satellite–ground laser communication links, atmospheric environmental factors have a significant impact on the transmission of optical signals, in which the change in the atmospheric wind speed directly affects the refractive index distribution of the atmospheric medium, which leads to the random disturbance of the propagation path of optical signals and changes in the phase front of the beam, and the existence of phase disturbances may lead to the distortion of optical signals and increase bit error rates. This,

in turn, reduces the communication quality and reliability, affecting the coherence and coupling efficiency of the light wave. The purpose of this study is to establish a simulation method based on the actual atmospheric environment by systematically analyzing the influence mechanism of the atmospheric wind speed on the phases of optical signals. In this method, the atmospheric wind speed, temperature, humidity, and other factors are comprehensively considered; the transmission process of optical signals in satellite–ground laser communication links is simulated through numerical calculations and simulation experiments, and the influence of wind speed changes on the phases of optical signals is analyzed.

Chen [1] developed a computational model to determine the average coupling efficiency of turbulence-perturbed space light in single-mode fibers when using annular aperture reception. This model employs a stepwise accumulation approach, facilitating easier computation and analysis. Chen and Yang [2] presented a new theoretical framework that takes into account the distinct nature of spatial mode reception, enabling a more precise characterization of the effect of eddy currents on optical signals. They identified key parameters that influence the evaluation of the correlation coefficient, such as the optical aperture Fresnel number, coherence Fresnel number, separation Fresnel number, and mode Fresnel number. Furthermore, they derived analytical asymptotic formulas for correlation coefficients, which facilitate the assessment of correlations with different parameters and provide valuable insights for practical applications. Traditional methods for the numerical modeling of light wave propagation under atmospheric turbulence conditions face limitations, particularly when dealing with the laser tracking of large target trajectories, adaptive optics, and the performance of atmospheric imaging optical systems. To address these issues, a novel approach is introduced that enables the generation of extended phase screens representing atmospheric-turbulence-induced phase distortion. This method also allows for the creation of long phase screens with spatially inhomogeneous statistical properties. By employing this approach, the performance of laser tracking and directional energy systems can be better modeled across a broad range of target trajectories [3–5]. We assessed the fiber coupling efficiency under atmospheric turbulence distortions and introduced a solution that employs a coherent fiber array serving as the receiver. This array rectifies phase distortions caused by atmospheric turbulence by adjusting the light phase in each sub-aperture, thus boosting the fiber coupling efficiency. Simulations indicate that coherent fiber arrays notably enhance the coupling efficiency, particularly under severe atmospheric turbulence conditions [6–8]. A previous study examined the correlation between the root-mean-square bandwidth of luminous flux time fluctuations in turbulent atmospheres and the position-dependent transverse wind speed. Additionally, it introduced the concept of the weight-averaged transverse wind speed. The findings revealed that under isotropic turbulence conditions, variations in the transverse wind speed primarily influence the root-mean-square bandwidth because of its magnitude rather than its direction [9]. Using the power spectrum inversion method, we constructed a turbulent phase screen to simulate the temporal variations in turbulence caused by atmospheric wind speed. We achieved this by implementing a transverse transitional phase screen. Furthermore, we conducted a detailed analysis of the time course of the laser transmission in atmospheric turbulence. Additionally, we quantitatively examined the impacts of the atmospheric wind speed on the phase characteristics of the laser beam, paying special attention to the wavefront phase's power spectral density. The results indicate that in boundary layer turbulence, the atmospheric wind speed exerts a relatively minor influence on the beam's phase characteristics. Conversely, in free-atmosphere turbulence, both the degree of the wavefront phase's aberration and the proportion of high-frequency phases tend to escalate with increasing atmospheric wind speed [10–13] came before [14]. Traditional methods for the numerical modeling of light wave propagation under atmospheric turbulence conditions face limitations, especially in assessing the performances of laser tracking, adaptive optics, and atmospheric imaging optical systems involving large-scale target trajectories. Vorontsov, A.M.; Paramonov, P.V.; and Valley, M.T. introduce a novel approach for generating ex-

tended phase screens that represent atmospheric-turbulence-induced phase distortions. This method also allows for the creation of long phase screens possessing spatially inhomogeneous statistical properties. This innovative technique enables more accurate modeling of laser tracking and directional energy systems' performances across a broad range of target trajectories [14,15]. Liu, T. and Zhang, J. [16,17] introduced a new simulation method rooted in optimization. Unlike the traditional sub-harmonic approach, this method does not rely on an equally spaced sampling of the low-frequency region. Instead, it determines the optimal distribution of sampling points through a searching algorithm, thereby minimizing the relative error between the desired and theoretical structure functions. Rezaee, M.; Rajabi, Y.; and Kokabi, K. [18] examined optical turbulence in the interstellar medium through simulations. They employed the phase screen simulation method to understand the propagation characteristics of light waves in this medium. Meanwhile, Yang, Z.Q.; Yang, L.H.; and Gong, L. [19] studied the distortion effects of atmospheric turbulence on Gaussian beam waveforms. Utilizing the phase screen method, they simulated the propagation of light waves in atmospheric turbulence, offering valuable insights into how atmospheric turbulence impacts beam transmission. Xiaoxin, Z.; Kai, H.; and Fuxing, F. focused on the phase properties of light waves propagating obliquely through a turbulent atmosphere. Using the phase screen simulation method, they uncovered the behavior of light waves under such conditions [20–30].

To better comprehend and address the impacts of atmospheric wind speed on the phases of optical signals in satellite–ground laser communication links, this paper introduces a Monte Carlo simulation method. This approach relies on the spatial and temporal variations in the atmosphere's refractive index, coupled with the principles of optical signal propagation in the atmosphere. It simulates how the phase of optical signals changes under varying wind speeds. This simulation allows for an in-depth exploration of how atmospheric wind speed affects the phases of optical signals. In the sections that follow, we will delve into the simulation method we have devised, along with its corresponding experiments and outcomes. Specifically, we will (1) examine the fundamental physics behind optical signal propagation in the atmosphere; (2) explore, under the coupling conditions present in satellite–ground laser communication links, how received optical signal fluctuations behave. Utilizing a Monte Carlo approach, we simulate a phase screen and a logarithmic amplitude screen. Additionally, we model the probability density of the received optical signal's power spectrum and the optical signal's time autocorrelation coefficient. (3) The model for the received signal's power probability density and the optical signal's time autocorrelation coefficient were verified using the cubic spline interpolation fitting method.

## 2. The Influence of the Atmospheric Wind Speed on Optical Wave Transmission in Atmospheric Random Channels

The satellite–ground laser communication link comprises two integral components: the uplink and the downlink. The uplink specifically refers to the connection established between a ground-based transmitting terminal and a space satellite. A defining characteristic of this uplink is the placement of an optical receiving terminal on a space platform, whether it be a low-orbit satellite or a space station. Conversely, the transmitting terminal resides within the atmosphere, close to Earth's surface. Upon entering the atmosphere, the laser beam is immediately subjected to refraction and attenuation. Moreover, atmospheric turbulence, characterized by fluctuations in the beam's angle of arrival and scintillation, further reduces the actual power received by the link receiver. Passing through the atmospheric channel significantly affects the quality of the laser beam, leading to significant deviations in its transmission direction and varying degrees of degradation. As the transmission beam traverses free space and reaches the receiving end, it experiences significant degradation, greatly compromising the quality of the link's optical signal system. More importantly, maintaining and achieving beam aiming, acquisition, and tracking functions at the micro-soliton level becomes an immense challenge. The downlink pertains to the

connection between the laser communication transmitter, positioned on the satellite in the upper atmosphere, and the ground receiver at the base station. During free space transmission, the light beam maintains its quality because of minimal deflection. However, upon entering the atmosphere, it encounters attenuation, turbulence, and deflection effects. Notably, the atmospheric channel's impact on the downlink is lesser compared to the uplink, making ground reception relatively easier.

Earth's atmosphere is filled with numerous gas molecules and atmospheric aerosol particles trapped by the planet's gravity. In a satellite–ground laser communication link, the laser beam must traverse the entire near-Earth atmosphere. During this journey, the laser interacts with these particles and molecules, giving rise to the laser atmosphere effect. This effect primarily impacts the amplitude and spatial phase of the optical field's transmitted wavefront. Amplitude effects manifest as random time variations in the optical field, including power loss, power fluctuations, and frequency filtering. Spatial effects, on the other hand, involve alterations in the beam's transmission direction, beam spreading, wavefront phase variations, and changes in phase coherence. The atmospheric wind speed influences the amplitude and phase characteristics of the laser beam transmitted through atmospheric turbulence. An increase in wind speed leads to a greater degree of wavefront amplitude and phase distortion. Both the amplitude and phase of the signal light are impacted by atmospheric turbulence, resulting in reduced fiber coupling efficiency [4,6].

### 3. Basic Model for Fiber Coupling

The process for coupling a space laser to a single-mode fiber in the downlink of satellite–ground laser communication is illustrated in Figure 1. The coupling efficiency ($\eta$) is the ratio of the optical power incident on the end face of the fiber to the total power emitted from the light source. The coupling efficiency of an acclimated Gaussian into a single-mode fiber through a single lens can be expressed as follows [1]:

$$\eta = \frac{|P_c|^2}{p_a} = \frac{\left| \iint_A U_i(r) U_m^*(r) d^2 r \right|^2}{\iint_A |U_i(r)|^2 d^2 r} \tag{1}$$

where $|r| \leq D/2$ is a point located on the pupil plane within the aperture, $D$ indicates the receiving lens's diameter, $P_c$ represents the mode coupling coefficient, $P_a$ represents the total optical power entering the receiving aperture, $U_i(r) = U_0 exp[\chi(r) + iS(r)]M(r)$ denotes the incident optical field transmitted at the aperture, $U_0(r)$ [2] is the optical field in the absence of turbulence [8–10], $\chi(r)$ is the amplitude fluctuations of the optical wave field, $S(r)$ describes the complex phase perturbation induced by atmospheric turbulence, $M(r)$ [2] is the aperture transmittance function, $U_m(r)$ [1] is the back-propagated fiber mode profile in the receiver aperture plane ($A$), and the asterisk indicates the complex conjugate. The mode coupling coefficient ($P_c$) can be written as follows [1,24]:

$$P_c = \iint_A U_0(r) \exp(\chi(r) + iS(r)) W(r) U_m^*(r) d^2 r \tag{2}$$

where $U_m(r)$ can be written as [1,24]

$$U_m(r) = \frac{\sqrt{2\pi} W_m}{\lambda f} \exp\left[ -\left( \frac{\pi W_m r}{\lambda f} \right)^2 \right] \tag{3}$$

where $W_m$ is the fiber mode field radius at the fiber end face, $f$ is the focal length of the receiver lens, and $\lambda$ is the optical wavelength. By substituting Equation (3) into Equation (2) [1,24],

$$P_c = \frac{\sqrt{2\pi} W_m}{\lambda f} \int_0^{D/2} \int_0^{D/2} U_0(r) \exp\left[ -\left( \frac{\pi W_m r}{\lambda f} \right)^2 \right] \times \exp[\chi(r) + iS(r)] d^2 r \tag{4}$$

Equation (4) can be reduced to [1,24]

$$P_c = \frac{\alpha \gamma r_0}{\sqrt{2\pi}} \int_0^1 \int_0^1 d^2\varepsilon U_0(\gamma r_0 \varepsilon/2) \exp\left(-\alpha^2 \varepsilon^2\right) \times \exp[\chi(\gamma r_0 \varepsilon/2) + iS(\gamma r_0 \varepsilon/2)] \quad (5)$$

$P_a$ in Equation (1) can be written as

$$P_a = \int_0^{D/2} \int_0^{D/2} |U_i(r)|^2 d^2r = \frac{\gamma^2 r_0^2}{4} \int_0^1 \int_0^1 \left|U_i\left(\frac{\gamma r_0 \varepsilon}{2}\right)\right|^2 d^2\varepsilon \quad (6)$$

where $r = D\varepsilon/2$, $\gamma = D/r_0$ denotes the turbulence intensity, $r_0$ denotes the atmospheric coherence length, and $\alpha = \pi D W_m/(2\lambda f)$ characterizes the coupling geometry. By substituting Equations (4) and (5) into Equation (1), the coupling efficiency expression obtained from the simplification is

$$\eta(\alpha, \gamma) = \frac{2\alpha^2 \left|\int_0^1 \int_0^1 d^2\varepsilon U_0(\gamma r_0 \varepsilon/2) \exp\left(-\alpha^2 \varepsilon^2\right) \times \exp[\chi(\gamma r_0 \varepsilon/2) + iS(\gamma r_0 \varepsilon/2)]\right|^2}{\pi \int_0^1 \int_0^1 |U_0(\gamma r_0 \varepsilon/2) \exp[\chi(\gamma r_0 \varepsilon/2) + iS(\gamma r_0 \varepsilon/2)|^2 d^2\varepsilon} \quad (7)$$

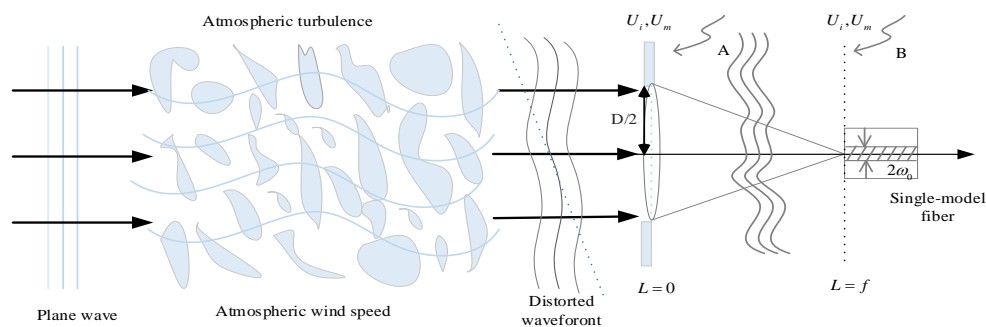

**Figure 1.** Schematic diagram illustrating the coupling of a space laser to a single-mode fiber in a satellite-to-ground laser communication downlink.

The random fields for the log-amplitude and phase fluctuations are respectively expressed as [2]

$$\chi(\mathbf{r}) = \sum_{n=1}^{M} f(p_n) \cos[\mathbf{p_n} \cdot (\mathbf{r} \text{-} \mathbf{v}t) + \mu_n],$$

$$S(\mathbf{r}) = \sum_{n=1}^{M} g(p_n) \cos[\mathbf{p_n} \cdot (\mathbf{r} \text{-} \mathbf{v}t) + \mu_n + v_n] \quad (8)$$

where $p_n$ is a random 2D wavenumber vector, and M denotes the number of wavenumber vector components, $\mu_n$ and $v_n$ are random angles, and $f(p_n)$ and $g(p_n)$ are random amplitudes associated with the cosine functions. The magnitude $p_n$ of $p_n$ is uniformly distributed in the range $[ln(k_0/k_n), ln(k_m/k_n)]$, where $k_n$ is an arbitrarily chosen normalization factor; $k_0 = 2\pi/L$, where $L$ is the outer scale of the turbulence and denotes the low-frequency cutoff wavenumber of the spatial power spectrum of refractive-index fluctuations; and $k_m = 2\pi/l_0$, where $l_0$ is the inner scale of the turbulence and represents the high-frequency cutoff wavenumber of the spatial power spectrum of the refractive-index fluctuations. The orientation angle of $p_n$ is uniformly distributed in the interval $[-\pi, \pi]$, $v = [v_x, v_y]$ indicates the transverse wind speed, $t$ is the light transmission time, $\mu_n$ is a random number that is uniformly distributed between $-\pi$ and $\pi$, and $v_n$ is a Gaussian-distributed random number with its probability density function (PDF) given by [2]

$$p_v(v_n) = \frac{1}{2\sqrt{\pi\alpha(p_n)}} \exp\left[-\frac{v_n^2}{4\alpha(p_n)}\right] \quad (9)$$

with

$$\alpha(p_n) = \ln[\Lambda(p_n)] \tag{10}$$

$$\Lambda(p_n) = \frac{\sqrt{\varphi_\chi(p_n)\varphi_S(p_n)}}{\varphi_{\chi S}(p_n)} \tag{11}$$

$$\varphi_\chi(p_n) = \begin{cases} C_1[q^2 - \sin(q^2)]q^{-17/3}, & k_0\sqrt{L/k} \le q \le k_m\sqrt{L/k}, \\ 0, & otherwise, \end{cases} \tag{12}$$

$$\varphi_S(p_n) = \begin{cases} C_1[q^2 + \sin(q^2)]q^{-17/3}, & k_0\sqrt{L/k} \le q \le k_m\sqrt{L/k}, \\ 0 & otherwise, \end{cases} \tag{13}$$

$$\varphi_{\chi S}(p_n) = \begin{cases} 2C_1\sin^2(q^2/2)q^{-17/3}, & k_0\sqrt{L/k} \le q \le k_m\sqrt{L/k}, \\ 0, & otherwise, \end{cases} \tag{14}$$

where $q = p_n(L/K)^{1/2}$, $C_1 = 1.536r_0^{-5/3}(L/k)^{11/6}$, and $r_0 = 2.1\rho$, and $r_0$ is typically referred to as Fried's atmospheric coherence width. It is commonly utilized to regulate the intensity of phase fluctuations in simulations of random phase screens; $\varphi_\chi(p_n)$ and $\varphi_S(p_n)$ are the Fourier transforms of the spatial correlation function of the log-amplitude and phase fluctuations; and $\varphi_{\chi S}(p_n)$ is the Fourier transform of the spatial cross-correlation function between the log-amplitude and phase fluctuations. It is noted that the variance in the Gaussian PDF given by Equation (9) is $2\alpha(p_n)$. The random amplitudes $f(p_n)$ and $g(p_n)$ are respectively defined by [2]

$$f(p_n) = \sqrt{\frac{\varphi_\chi(p_n)}{\pi M\Omega(p_n)}} \tag{15}$$

$$g(p_n) = \sqrt{\frac{\varphi_S(p_n)}{\pi M\Omega(p_n)}} \tag{16}$$

where

$$\Omega(p_n) = \frac{1}{2\pi p_n^2 \ln(k_m/k_0)} \tag{17}$$

According to the Kolmogorov turbulence statistical theory model, the spatial coherence radius of atmospheric turbulence in satellite–ground laser communication downlinks is derived as follows [3]:

$$\rho_0 = \left(\frac{\cos\zeta}{1.45\mu_0 k^2}\right)^{3/5} \tag{18}$$

where

$$\mu_0 = \int_{h_0}^{H} C_n^2(h)dh \tag{19}$$

where $H = h_0 + L\cos(\zeta)$ is the satellite altitude, $\zeta$ is the zenith angle, $h_0$ is the altitude at which the ground-receiving terminal is located, $L$ is the optical transmission distance, $k = 2\pi/\lambda$ is optical wavenumber, $C_n^2$ is a series of structural constants of the refractive index of the atmospheric turbulence satisfying Kolmogorov's statistical theory of turbulence.

The selection of the Hufnagel–Valley model for atmospheric turbulence in satellite–ground laser communication applications is preferred because of its simplicity and versatility. There are several reasons why it is often chosen over other turbulence models. (1) Empirical validity: The H-V model has been extensively validated using experimental data and observations under various atmospheric conditions. It provides a good approximation of turbulence effects in practical scenarios, such as satellite–ground laser communication and optical systems. (2) Simplified representation: Compared to more complex turbulence models, like the Kolmogorov or von Kármán models, the H-V model offers a simplified representation of atmospheric turbulence. It assumes a single-layered, isotropic, and homogeneous structure, making it easier to implement in simulations and analytical calculations.

(3) Computational efficiency: The simplicity of the H-V model leads to computational efficiency, making it suitable for real-time simulations and system-level analysis. Other turbulence models that consider finer details or specific atmospheric characteristics may require more computational resources and time. (4) Broad applicability: The versatility of the H-V model allows it to be applied in various fields, including wireless communications, radar systems, and astronomy. It provides a reasonable estimate of turbulence-induced effects on signal propagation, such as scintillation, fading, and beam spreading.

For slant transmission, the refractive index's atmospheric structure constant in the Hufnagel–Valley model is expressed as follows [3,9]:

$$C_n^2(h) = 5.94 \times 10^{-53} \left(\frac{w}{27}\right)^2 h^{10} \exp(-\frac{h}{1000}) + 2.7 \times 10^{-16} \exp(-\frac{h}{1500}) + A \exp(-\frac{h}{100}) \tag{20}$$

where h is in meters (m), w is the mean-square value of the transverse wind speed in meters per second (m/s), and $A = 1.7 \times 10^{-14}$ $m^{-2/3}$.

The mean-square value of the transverse wind speed in (20) is determined from

$$w = \left[\frac{1}{15 \times 10^3} \int_{5 \times 10^3}^{20 \times 10^3} V^2(h) dh\right]^{1/2} \tag{21}$$

where $V(h)$ is frequently characterized using the Bufton wind model, which can be expressed as follows:

$$V(h) = \omega_s h + V_g + 30 \exp\left[-\left(\frac{h - 9400}{4800}\right)^2\right] \tag{22}$$

The variable $V_g$ represents the ground wind speed, while $\omega_s$ denotes the slew rate that corresponds to the movement of a satellite relative to an observer on the ground.

## 4. Monte Carlo Simulation Results and Discussion

The phase screen and amplitude of the turbulence are simulated using hot wind flowing transversely through the beam during satellite–ground laser channel coupling. In simulating laser propagation in the atmosphere, including its time course, the phase and logarithmic amplitude of turbulence are employed. Herein, the atmospheric wind speed in turbulence is decomposed into two components: one parallel and the other perpendicular to the propagation direction. This decomposition is necessary because the laser's propagation speed vastly exceeds the atmospheric wind speed along the propagation path. The parallel component of the atmospheric wind speed minimally affects the simulation results of the turbulent phase and logarithmic amplitude in the transport process. Consequently, in simulating laser atmospheric propagation, we primarily consider the vertical component—the changes in the turbulent structure induced by the transverse wind speed. In the numerical simulation that encompasses the time course of the laser propagation in the atmosphere, the influence of the transverse wind speed on the turbulence can be replicated by constructing turbulent phase and logarithmic amplitude screens larger than the beam aperture and then shifting these screens laterally. This approach has been documented in references [4–6,12].

In the Monte Carlo simulation experiment conducted in this article, we utilized the H-V model, with the system model parameters outlined in Table 1. The length of the signal's sample sequence was set at $2 \times 10^4$, with a sampling time interval of 0.001 s [3–5,22]. Because only a limited number of samples can be used in the simulation experiment, there may be some errors between the correlation of the random sample signal and the theoretical correlation. To solve this problem, 200 simulation experiments were repeated, and the normalized autocorrelation function coefficients were calculated according to the results.

**Table 1.** The system model parameters.

| A | $L_0$ | $l_0$ | L | h | $\zeta$ | D | f | $\lambda$ | $\omega_0$ | $W_m$ |
|---|---|---|---|---|---|---|---|---|---|---|
| $1.7 \times 10^{-14}$ m$^{-2/3}$ | 1 m | 0.1 m | 300,000 m | 2000 m | 45 °C/180 | 0.25 m | 0.6 m | $1.55 \times 10^{-6}$ m | 0.1 m | $5.25 \times 10^{-6}$ m |

Figure 2 shows an analysis of the optical signal's transmission performance when the mean-square value of the transverse wind speed (w) = 21 m/s, the atmospheric coherence length ($r_0$) = 0.6035 m, and the transverse wind speed (v) = 1 m/s, v = 2 m/s, and v= 3 m/s. To illustrate the effectiveness of the proposed method, another set of data is used for the Monte Carlo simulation. Figure 3 shows an analysis of the optical signal's transmission performance when the mean-square value of the transverse wind speed (w) = 30 m/s, the atmospheric coherence length ($r_0$) = 0.46 m, and the transverse wind speed (v) = 1 m/s, v = 2 m/s, and v = 3 m/s. Through Figures 2a–f and 3a–f, it is found that the increase in the wind speed causes the fluctuation acceleration of the optical signal's power spectrum; it affects the model for the power spectral probability density of the optical signal, resulting in a reduction in the temporal auto correlation coefficient of the optical signal, and it affects the coupling efficiency.

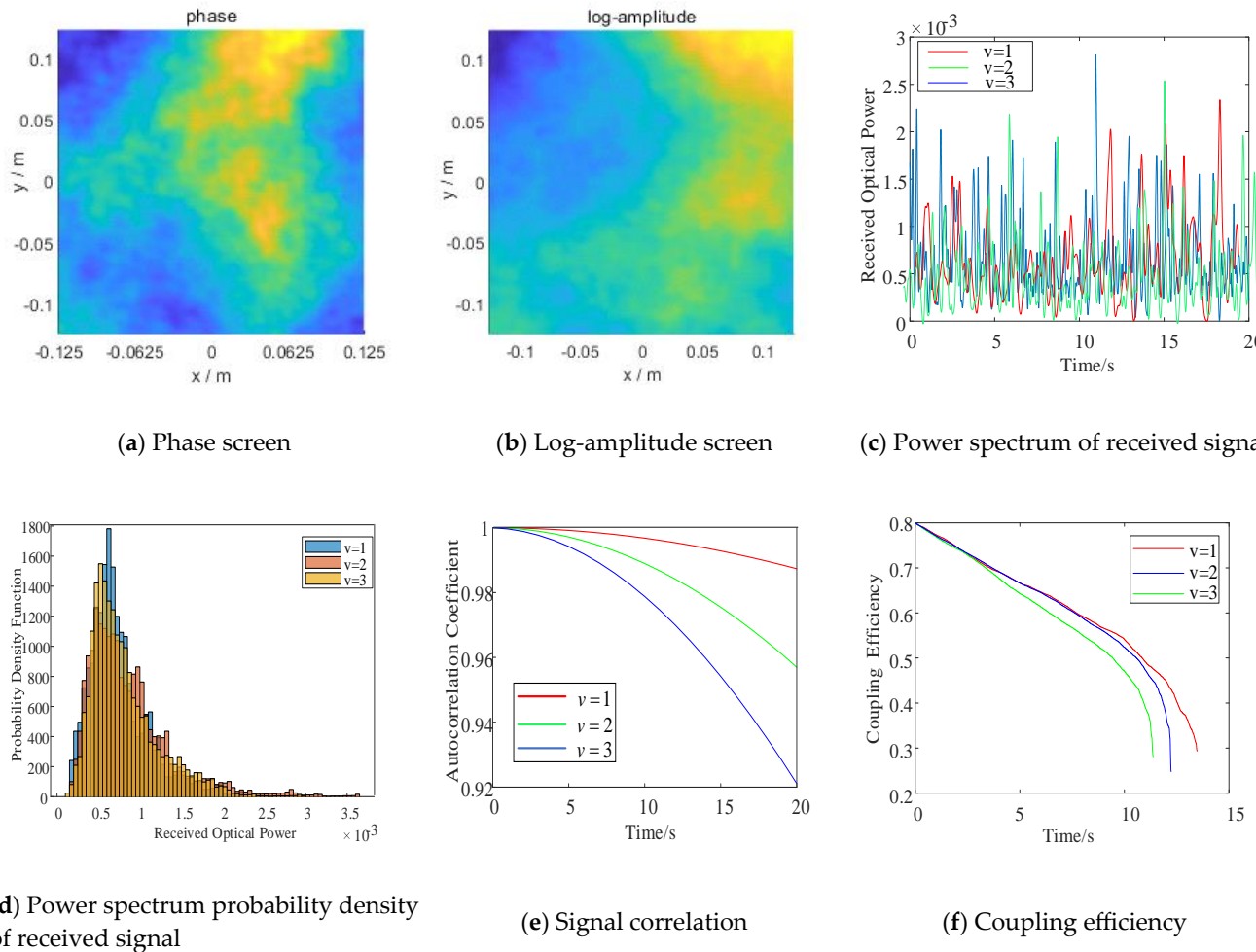

(**a**) Phase screen

(**b**) Log-amplitude screen

(**c**) Power spectrum of received signal

(**d**) Power spectrum probability density of received signal

(**e**) Signal correlation

(**f**) Coupling efficiency

**Figure 2.** Analysis of optical signal's transmission performance when the mean-square value of the transverse wind speed (w) = 21 m/s, the atmospheric coherence length ($r_0$) = 0.6035 m, and the transverse wind speed (v) = 1 m/s, v = 2 m/s, and v = 3 m/s, respectively. (**a**) Phase screen, (**b**) logarithmic amplitude screen, (**c**) change in the power spectrum of the optical signal, (**d**) change in the probability density of the power spectrum of the optical signal, (**e**) change in the correlation of the optical signal, and (**f**) change in the coupling rate.

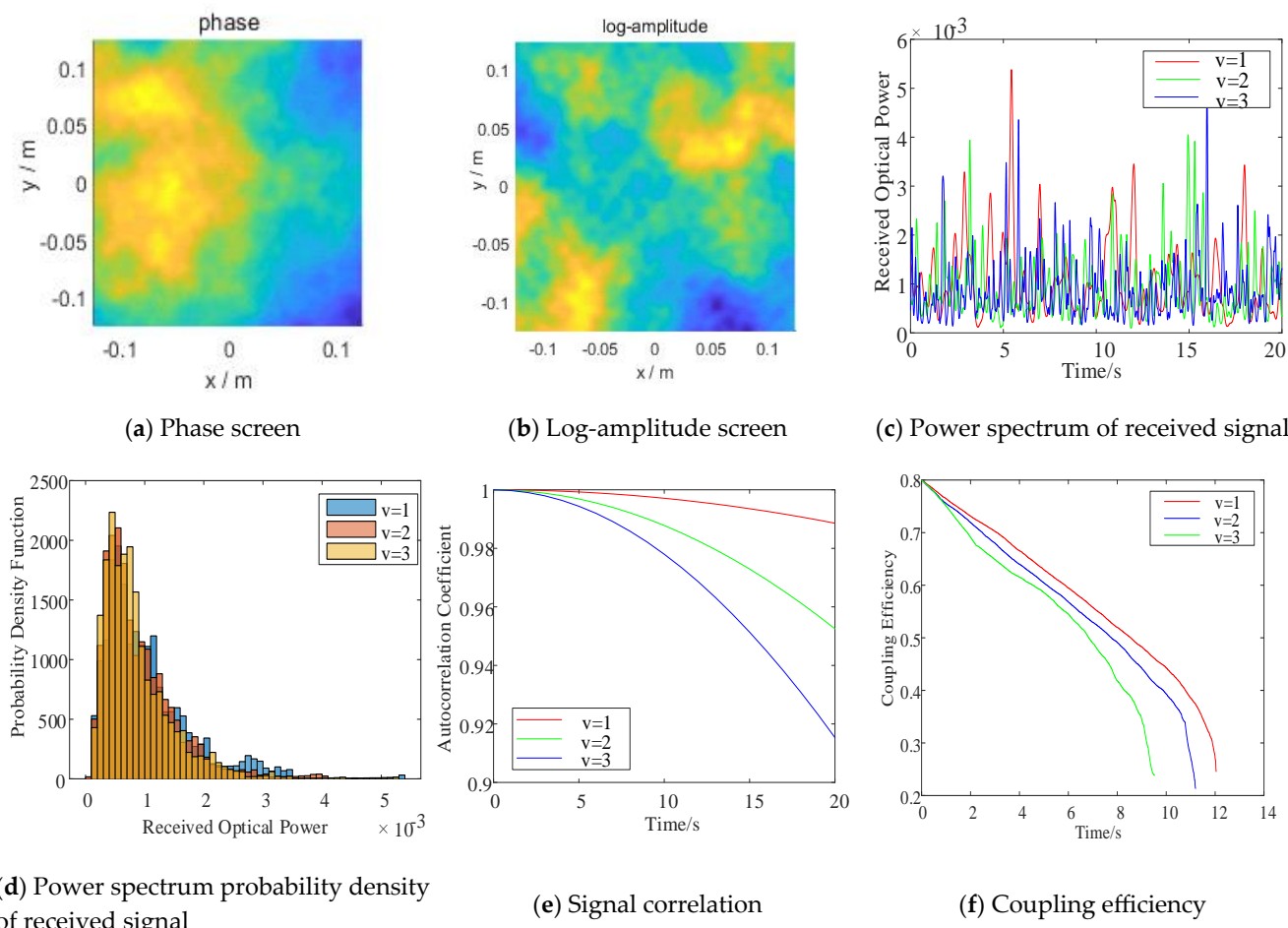

(**a**) Phase screen     (**b**) Log-amplitude screen     (**c**) Power spectrum of received signal

(**d**) Power spectrum probability density of received signal     (**e**) Signal correlation     (**f**) Coupling efficiency

**Figure 3.** Analysis of optical signal's transmission performance when the mean-square value of the transverse wind speed (w) = 30 m/s, the atmospheric coherence length ($r_0$) = 0.46 m, and the transverse wind speed (v) = 1 m/s, v = 2 m/s, and v = 3 m/s. (**a**) Phase screen, (**b**) logarithmic amplitude screen, (**c**) change in the power spectrum of the optical signal, (**d**) change in the probability density of the power spectrum of the optical signal, (**e**) change in the correlation of the optical signal, and (**f**) change in the coupling rate.

To verify the model for the probability density of the received signal's power spectrum and the model of the time autocorrelation coefficient, this paper uses the cubic spline interpolation method to fit the model for the general probability density of the received signal's power and the model for the time auto correlation coefficient. Figure 4 shows the fitting results for different transverse wind speeds, v = 1 m/s, v = 2 m/s, and v = 3 m/s, with a mean-square value of the transverse wind speed (w) = 21 m/s and an atmospheric coherence length ($r_0$) = 0.6035 m. Figure 5 shows the fitting results for different transverse wind speeds, v = 1 m/s, v = 2 m/s, and v = 3 m/s, with a mean-square value of the transverse wind speed (w) = 30 m/s and an atmospheric coherence length ($r_0$) = 0.46 m. The fitting results of the model for the time auto correlation coefficient of the optical signal are consistent, and the fitting results are as follows: Cubic spline interpolation: f(x) = piecewise polynomial computed from p, where x is normalized by mean 10 and Std. 6.205. Coefficients: p = coefficient structure; goodness of fit: SSE: 0; R-squared: 1; adjusted R-squared: Nan; RMSE: Nan. However, the fitting results of the probability density for the power spectrum of the received optical signal are different according to the different parameters. The fitting results are as follows: Cubic spline interpolation: f(x) = piecewise polynomial computed from p, where x is normalized by mean and Std. Table 2. Coefficients: p = coefficient structure; goodness of fit: SSE: 0; R-squared: 1; adjusted R-squared: Nan; RMSE: Nan. The analysis in Table 2 reveals that under the condition

of equal mean crosswind speeds, the fitted mean and standard deviation also vary with the change in the crosswind speed. It is observed from Figures 4 and 5 that the optical signal's temporal autocorrelation coefficient model and the received optical signal's power spectrum probability density model simulated by the Hufnagel–Valley model are basically consistent with the fitted model, indicating that the Hufnagel–Valley model can accurately represent the turbulence effect at different wind speeds.

**Table 2.** The data corresponding to the means and standard deviations used for fitting the received optical signal's power probability density under different wind speed conditions.

| $r_0$ | w | Mean | | | Std. | | |
|---|---|---|---|---|---|---|---|
| | | v = 1 m/s | v = 2 m/s | v = 3 m/s | v = 1 m/s | v = 2 m/s | v = 3 m/s |
| 0.6035 | 21 m/s | 0.001225 | 0.001925 | 0.001575 | 0.0006134 | 0.001018 | 0.0008443 |
| 0.460 | 30 m/s | 0.0028 | 0.0021 | 0.00275 | 0.001544 | 0.001198 | 0.001515 |

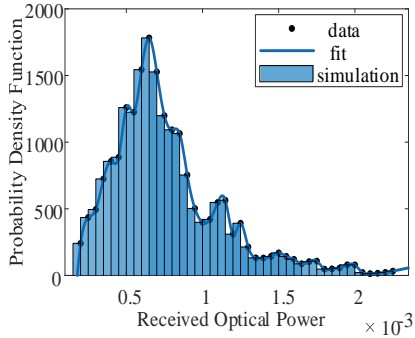 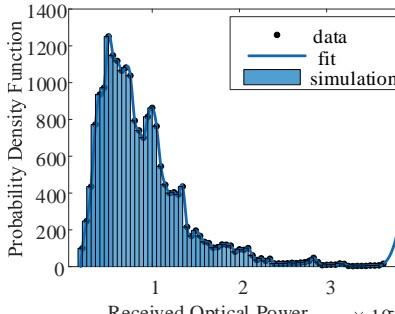 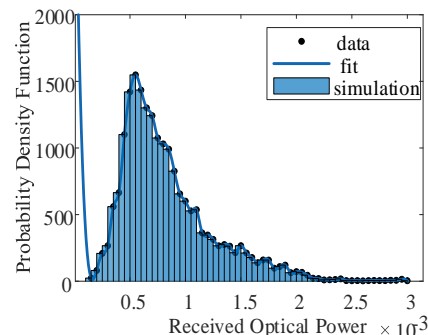

(**a**) Fit of the probability density model with v= 1 m/s    (**b**) Fit of the probability density model with v = 2 m/s    (**c**) Fit of the probability density model with v = 3 m/s

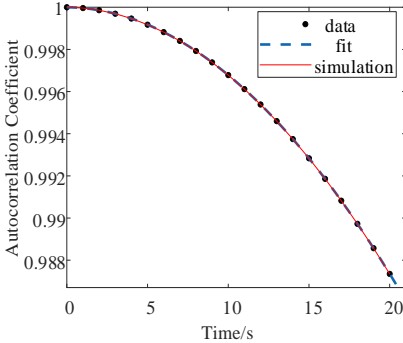 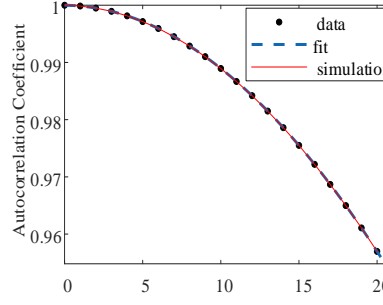 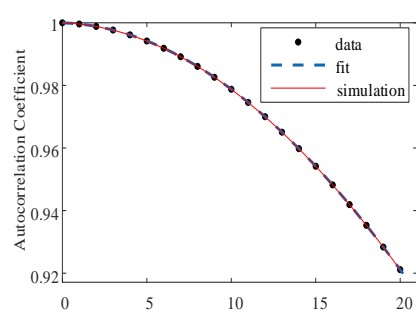

(**d**) Fit of the v = 1 m/s correlation model    (**e**) Fit of the v = 2 m/s correlation model    (**f**) Fit of the v = 3 m/s correlation model

**Figure 4.** Fits of received optical signal's power spectrum probability density model and fits of optical signal's auto correlation coefficient model when transverse wind speed's mean-square value (w) = 21 m/s, atmospheric coherence length ($r_0$) = 0.6035 m, and transverse wind speed (v) = 1 m/s, v = 2 m/s, and v= 3 m/s.

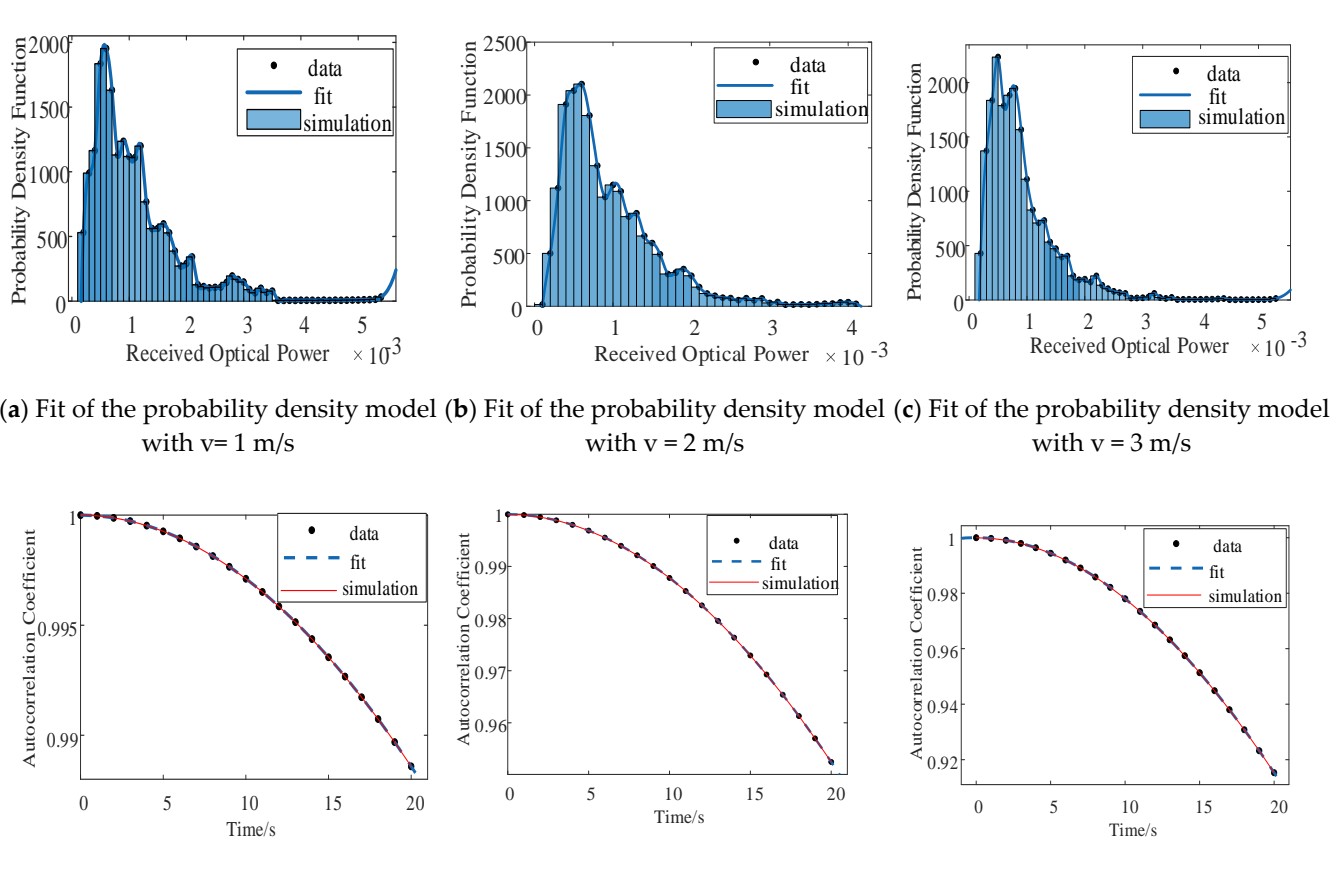

(**a**) Fit of the probability density model with v= 1 m/s

(**b**) Fit of the probability density model with v = 2 m/s

(**c**) Fit of the probability density model with v = 3 m/s

(**d**) Fit of the v = 1 m/s correlation model

(**e**) Fit of the v = 2 m/s correlation model

(**f**) Fit of the v = 3 m/s correlation model

**Figure 5.** Fits of received optical signal's power spectrum probability density model and fits of optical signal's auto correlation coefficient model when transverse wind speed's mean-square value (w) = 30 m/s, atmospheric coherence length ($r_0$) = 0.46 m, and transverse wind speed (v) = 1 m/s, v = 2 m/s, and v = 3 m/s, respectively.

## 5. Conclusions

In this paper, we simulate the transmission process of optical signals in satellite–ground laser communication links through numerical calculations and simulation experiments, focusing on the coupling efficiency. Additionally, we analyze how changes in wind speed affect the phases of optical signals. Our findings reveal that the atmospheric wind speed alters the amplitude and phase characteristics of laser beam propagation within atmospheric turbulence. Increasing the wind speed often intensifies the atmospheric turbulence, which leads to changes in the refractive index. These changes, in turn, alter the phase and amplitude of the light beam, causing it to deviate from its predetermined path. This deviation disrupts the directional transmission of the optical signal, alters the shape of the power spectrum's probability density model, reduces the time autocorrelation coefficient of the optical signal, and ultimately decreases the coupling efficiency, thereby affecting the reliability of the communication link. In this paper, the effectiveness of the proposed method in satellite–ground laser communication is verified through fitting analysis, providing strong support for turbulence coupling-efficiency studies. However, this method still faces limitations regarding real-time application, stability, and performance verification. In the future, we should delve deeper into the mechanisms of atmospheric turbulence's influence. Additionally, we need to develop cutting-edge optical technology and beam control methods and explore multi-beam parallel transmission to propel the continued advancement of laser communication technology.

**Author Contributions:** W.Z., formal analysis, validation, and writing; C.C., supervision and project administration. All authors have read and agreed to the published version of the manuscript.

**Funding:** This research received no external funding.

**Institutional Review Board Statement:** Not applicable.

**Informed Consent Statement:** Not applicable.

**Data Availability Statement:** The data are contained within the article.

**Acknowledgments:** This experimental work has been approved by the Computer, 3D Graphics, and Simulation Laboratory of the College of Computer Science and Technology at Changchun University of Science and Technology.

**Conflicts of Interest:** The authors declare no conflicts of interest.

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
