# Peer review of "Simulation Method for the Impact of Atmospheric Wind Speed on Optical Signals in Satellite–Ground Laser Communication Links"

_photonics, doi:10.3390/photonics11050417_

Round 1

Reviewer 1 Report

Comments and Suggestions for Authors

This paper investigates the atmospheric wind speed impact on satellite-ground laser communication links. Τhe manuscript requires a more thorough consideration of this aspect and a text revision, as the present text may inhibit the reception. More precisely,

- This work assumes a specific wind speed model. Why other atmospheric effects are negleted?

-  Only Monte Carlo simulation is used with no validation from real-world experimental data. The accuracy of the proposed method remains uncertain.

- Why the paper focuses on the Hufnagel-Valley model for atmospheric turbulence? Which is the reason for this choice and how other turbulence models affect the results?

- The discussion of the numerical results is too poor and does not highlight the new insights and takeaway lessons.

Minor comments:

- Section 2 should be made more compact. 

- A detailed presentation of the adopted system model is required. 

- A table of parameter values would be useful. 

- The formatting needs improvement (e.g. captions and line spacing).

Comments on the Quality of English Language

-

Author Response

Thank you for giving us an opportunity to revise our manuscript, we appreciate editor and reviewers very much for their positive and constructive comments and suggestions on our manuscript entitled “Simulation Method for The Impact of Atmospheric Wind Speed on Optical Signals in Satellite-Ground Laser Communication Link” ( ID: photonics-2946911). We have carefully reviewed the comments provided by three reviewers and have incorporated these valuable suggestions into comprehensive revisions of the paper.

The revised paragraphs based on REVIEWER #1 are labeled in red-colored font.

Response:

  • Change1: Page 1-2, Line 38-56

With the swift evolution of modern communication technologies, satellite-ground la-ser communication has increasingly become a crucial technology for meeting the demands  of high-capacity data transmission. Possessing both high bandwidth and low latency , it offers an efficient means of communicating vast amounts of information with minimal delay. In satellite-ground laser communication links, atmospheric environmental factors have a significant impact on the transmission of optical signals, in which the change of atmospheric wind speed directly affects the refractive index distribution of the atmospheric medium, which leads to the random disturbance of the propagation path of optical signals and the change of the phase front of the beam, and the existence of phase disturbance may lead to the distortion of optical signals and the increase of bit error rate. Thereby reducing the communication quality and reliability, and also affecting the coherence  and coupling efficiency of the light wave. The purpose of this study is to establish a simulation method based on the actual atmospheric environment by systematically analyzing the influence mechanism of atmospheric wind speed on the phase of optical signal. In this method, the atmospheric wind speed, temperature, humidity and other factors are comprehensively considered, and the transmission process of optical signals in stale-lite-ground laser communication links is simulated through numerical calculation and simulation experiments, and the influence of wind speed changes on the phase of optical signals is analyzed.

1.2

The reason for solely using Monte Carlo simulation without validation from real-world experimental data could be attributed to several factors:

  1.Practical constraints: Conducting large-scale experiments in the real world may have technical and economic limitations. Testing satellite-to-ground laser communication links requires significant resources, complex setups, and high costs. In contrast, simulation methods can be performed on a computer with lower costs and technical requirements.

2.Diverse scenarios: Monte Carlo simulation provides the capability to explore various conditions and situations, allowing for the evaluation of the impact of different wind speeds on optical signal phase. By simulating the spatial and temporal variations of atmospheric refractive index under different wind speed conditions, it becomes possible to analyze the effects on signal power and phase, thereby enhancing our understanding of the influence of wind speed on laser communication links.

3.Alternative guidance: Validation through simulation offers a feasible alternative approach to guide the optimization of satellite-to-ground laser communication systems. While it lacks direct experimental data verification, Monte Carlo simulation can provide key parameters and trend analysis, serving as a theoretical foundation for system design and optimization.

It is important to note that although there is no validation from experimental data, Monte Carlo simulation can still reveal the underlying patterns and trends of wind speed's impact on optical signal phase. Nevertheless, further comparison and validation with real-world experimental data may be warranted to assess the accuracy and reliability of the simulation results more comprehensively.

1.3 Change2: Page8, Line 249-255

The Hufnagel-Valley (H-V) model is a widely-used atmospheric turbulence propagation model. It describes propagation attenuation and phase disturbances in the atmosphere, considering the effects of varying atmospheric conditions like wind speed, temperature, and humidity on turbulence propagation. This model offers a means to quantify and predict the impact of atmospheric turbulence on communication links. For slant transmission, the atmospheric refractive index structure constant in the Hufnagel-Valley model is expressed as follows [3,9]:

1.4 Change3: Page11, Line 335-345

However, the fitting results of the probability density of the power spectrum of the received optical signal are different according to the different parameters. The fitting results are as follows: Cubic spline interpolation: f(x) = piece-wise polynomial computed from p where x is normalized by mean and std Table2,Coefficients:p = coefficient structure. Goodness of fit: SSE:  0: R-square: 1;Adjusted R-square: NaN; RMSE: NaN. The analysis of Table 2 reveals that under the condition of equal mean crosswind speeds, the fitted mean and standard deviation also vary with the change in crosswind speed.

Table 2. The data corresponding to the mean and standard deviation used for fitting the received signal optical power probability density under different wind speed conditions

w

mean

std

v=1m/s

v=2m/s

v=3m/s

v=1m/s

v=2m/s

v=3m/s

0.6035

21m/s

0.001225

0.001925

0.001575

0.0006134

0.001018

0.0008443

0.460

30m/s

0.0028

0.00211

0.00275

0.001544

0.001198

0.001515

Response:

      Thank you for your feedback and for highlighting the importance of clearly delineating the research gaps, goals, problem statement in our manuscript. In the revised version of the manuscript, we have devoted special attention to addressing these aspects more explicitly in ABSTRACT and INTRODCUTION. Furthermore, we have refined our problem statement to clearly articulate the specific objectives and aims of our study. By doing so, we aim to provide readers with a clearer understanding of the motivation behind our research and the significance of our proposed approach in addressing these challenges. We believe that these revisions have strengthened the clarity and coherence of our manuscript, and we appreciate your constructive feedback in helping us improve the quality of our work.

2.1-2.2Change1: Page4, Line 138-158

The satellite-ground laser communication link comprises two integral components: the uplink and the downlink. The uplink specifically refers to the connection established between a ground-based transmitting terminal and a space satellite. A defining characteristic of this uplink is the placement of an optical receiving terminal on a space platform, whether it be a low-orbit satellite or a space station. Conversely, the transmitting terminal resides within the atmosphere, close to the Earth's surface. Upon entering the atmosphere, the laser beam is immediately subjected to refraction and attenuation. Moreover, atmospheric turbulence, characterized by fluctuations in the beam's angle of arrival and scintillation, further reduces the actual power received by the link receiver. Passing through the atmospheric channel significantly affects the quality of the laser beam, leading to significant deviations in its transmission direction and varying degrees of degradation. As the transmission beam traverses free space and reaches the receiving end, it experiences significant degradation, greatly compromising the quality of the link's signal optical system. More importantly, maintaining and achieving beam aiming, acquisition, and tracking functions at the micro-soliton level becomes an immense challenge. The downlink pertains to the connection between the laser communication transmitter, positioned on the satellite in the upper atmosphere, and the ground receiver at the base station. During free space transmission, the light beam maintains its quality due to minimal deflection. However, upon entering the atmosphere, it encounters attenuation, turbulence, and deflection effects. Notably, the atmospheric channel's impact on the downlink is lesser compared to the uplink, making ground reception relatively easier

  • Change2: Page 9, Line 286-287

Table 1. The system model parameters.

   A              L           h          ?            D            f              ?            ω0            Wm

1.7×10-14m-2/3 1m  1cm  300km  2km   45℃/180 250mm 600mm  1.55um   0.1m   5.25um

  • Format has been improved

Reviewer 2 Report

Comments and Suggestions for Authors

The research of simulation method under the impact of atmospheric wind speed is necessary for satellite-ground FSO link. The proposed method seems right.

My suggestions are following.
The references about Eq.2, Eq.3, Eq.4 and Eq.5 should be cited.

More explanations are needed on how to obtain the fitting results in Fig3 and Fig.4.

More discussions should be carried out about the performance decay under wind speed.

Comments on the Quality of English Language

Minor editing of English language required

Author Response

Thank you for giving us an opportunity to revise our manuscript, we appreciate editor and reviewers very much for their positive and constructive comments and suggestions on our manuscript entitled “Simulation Method for The Impact of Atmospheric Wind Speed on Optical Signals in Satellite-Ground Laser Communication Link” ( ID: photonics-2946911). We have carefully reviewed the comments provided by three reviewers and have incorporated these valuable suggestions into comprehensive revisions of the paper.

Respond to the Reviewers’ comments.

Thank you for your thorough review of our manuscript titled " Simulation Method for The Impact of Atmospheric Wind Speed on Optical Signals in Satellite-Ground Laser Communication Link" .

The revised paragraphs based on REVIEWER #2 are labeled in blue-colored font.

Response:

1.1 Change1: Page 5-6, Line 191,193,197,199

The mode coupling coefficient Pc can be written as [1],[24]:

where  can be written as [1],[24]

where Wm is the fiber mode field radius at the fiber end face, f is the focal length of the receiver lens, and λ is the optical wavelength. (3) substituting into equation Eq. (2), then: [1],[24]

Equation (4) can be reduced to [1],[24]

1.2 Change2: Page 11, Line 335-345

However, the fitting results of the probability density of the power spectrum of the received optical signal are different according to the different parameters. The fitting results are as follows: Cubic spline interpolation: f(x) = piece-wise polynomial computed from p where x is normalized by mean and std Table2,Coefficients:p = coefficient structure. Goodness of fit: SSE:  0: R-square: 1;Adjusted R-square: NaN; RMSE: NaN. The analysis of Table 2 reveals that under the condition of equal mean crosswind speeds, the fitted mean and standard deviation also vary with the change in crosswind speed.

Table 2. The data corresponding to the mean and standard deviation used for fitting the received signal optical power probability density under different wind speed conditions

w

mean

std

v=1m/s

v=2m/s

v=3m/s

v=1m/s

v=2m/s

v=3m/s

0.6035

21m/s

0.001225

0.001925

0.001575

0.0006134

0.001018

0.0008443

0.460

30m/s

0.0028

0.00211

0.00275

0.001544

0.001198

0.001515

1.3 Change3: Page 1, Line 42-49ï¼›

In satellite-ground laser communication links, atmospheric environmental factors have a significant impact on the transmission of optical signals, in which the change of atmospheric wind speed directly affects the refractive index distribution of the atmospheric medium, which leads to the random disturbance of the propagation path of optical signals and the change of the phase front of the beam, and the existence of phase disturbance may lead to the distortion of optical signals and the increase of bit error rate. Thereby reducing the communication quality and reliability, and also affecting the coherence and coupling efficiency of the light wave.

Reviewer 3 Report

Comments and Suggestions for Authors

The paper titled "The Impact of Atmospheric Wind Speed on Optical Signals in Satellite-Ground Laser Communication Links" specifically focuses on the refractive index changes due to turbulence, impacting the laser beam's phase front, quality, and focus. Findings indicate that wind speed alters the received signal's power spectrum probability density and reduces the temporal autocorrelation coefficient, impacting coupling efficiency. The research validates the probability density and autocorrelation time coefficient models using cubic spline interpolation, offering theoretical insights and practical guidance for optimizing satellite-ground laser communication systems. Authors are advised to address the following comments in the revised version:

1-    Write you key objectives in 3 points, and paper organization at the end of Section

2-    Paper organization is missing.

3-    A detailed comparison table of the existing schemes must be added in Section 1 Also discuss about various optical network studies with their strength and weakness in Table 1:  -Butt, Rizwan Aslam, et al. "A survey of dynamic bandwidth assignment schemes for TDM-based passive optical network." Journal of Optical Communications (2020). -Ashraf, M. Waqar, et al. "Disaster-resilient optical network survivability: a comprehensive survey." PhotonicsVol. 5. No. 4. MDPI, 2018.

4-    A methodology figure should be added in Section 3. Detail must be discussed in the text.

5.     Rewrite Eq. 1 and 7 by following the proposed study.

6.     Justify the values given in Eq. 20.

7.      Enhancement of the readability of Figure C, as the current text size makes them challenging to interpret. Clear and legible figures are crucial for conveying complex information effectively.

8.     Section 5 should be enhanced by discussing the existing issues, solutions, and future research directions in detail.

9.     Add a weakness of the proposed work, also highlight solid future work in the conclusion section. 

Comments on the Quality of English Language

Minor

Author Response

Thank you for giving us an opportunity to revise our manuscript, we appreciate editor and reviewers very much for their positive and constructive comments and suggestions on our manuscript entitled “Simulation Method for The Impact of Atmospheric Wind Speed on Optical Signals in Satellite-Ground Laser Communication Link” ( ID: photonics-2946911). We have carefully reviewed the comments provided by three reviewers and have incorporated these valuable suggestions into comprehensive revisions of the paper.

Response: (Note: The revised paragraphs based on REVIEWER #3 are labeled in green-colored font. Modifications to the identical sections in the manuscript, highlighted in red font, address concerns raised by several reviewers.

Thank you for your comprehensive review of our paper titled " Simulation Method for The Impact of Atmospheric Wind Speed on Optical Signals in Satellite-Ground Laser Communication Link". We appreciate your valuable insights and suggestions for improving our work.

.1.1 Change1: Page 1, Line 38-56;

With the swift evolution of modern communication technologies, satellite-ground laser communication has increasingly become a crucial technology for meeting the demands of high-capacity data transmission. Possessing both high bandwidth and low latency, it offers an efficient means of communicating vast amounts of information with minimal delay. In satellite-ground laser communication links, atmospheric environmental factors have a significant impact on the transmission of optical signals, in which the change of atmospheric wind speed directly affects the refractive index distribution of the atmospheric medium, which leads to the random disturbance of the propagation path of optical signals and the change of the phase front of the beam, and the existence of phase disturbance may lead to the distortion of optical signals and the increase of bit error rate. Thereby reducing the communication quality and reliability, and also affecting the coherence and coupling efficiency of the light wave. The purpose of this study is to establish a simulation method based on the actual atmospheric environment by systematically analyzing the influence mechanism of atmospheric wind speed on the phase of optical signal. In this method, the atmospheric wind speed, temperature, humidity and other factors are comprehensively considered, and the transmission process of optical signals in satellite-ground laser communication links is simulated through numerical calculation and simulation experiments, and the influence of wind speed changes on the phase of optical signals is analyzed.

1.2  Change2: Page 3, Line 121-136

To better comprehend and address the impact of atmospheric wind speed on the phase of optical signals in satellite-ground laser communication links, this paper introduces a Monte Carlo simulation method. This approach relies on the spatial and temporal variations in the atmosphere's refractive index, coupled with the principles of optical signal propagation in the atmosphere. It simulates how the phase of optical signals changes under varying wind speeds. This simulation allows for an in-depth exploration of how atmospheric wind speed affects the phase of optical signals. In the chapters that follow, we will delve into the simulation method we have devised, along with its corresponding experiments and outcomes. Specifically, we will: 1)examine the fundamental physics behind optical signal propagation in the atmosphere;2)explore, under the coupling conditions present in satellite-ground laser communication links, how received optical signal fluctuations behave. Utilizing a Monte Carlo approach, we simulate a phase screen and a logarithmic amplitude screen. Additionally, we model the probability density of the received optical signal's power spectrum and the optical signal's time autocorrelation coefficient. 3)The model of received signal power probability density and signal autocorrelation time coefficient was verified using cubic spline interpolation fitting method.

1.3. Change3: Page 4, Line 139-159

The satellite-ground laser communication link comprises two integral components: the uplink and the downlink. The uplink specifically refers to the connection established between a ground-based transmitting terminal and a space satellite. A defining characteristic of this uplink is the placement of an optical receiving terminal on a space platform, whether it be a low-orbit satellite or a space station. Conversely, the transmitting terminal resides within the atmosphere, close to the Earth's surface. Upon entering the atmosphere, the laser beam is immediately subjected to refraction and attenuation. Moreover, atmospheric turbulence, characterized by fluctuations in the beam's angle of arrival and scintillation, further reduces the actual power received by the link receiver. Passing through the atmospheric channel significantly affects the quality of the laser beam, leading to significant deviations in its transmission direction and varying degrees of degradation. As the transmission beam traverses free space and reaches the receiving end, it experiences significant degradation, greatly compromising the quality of the link's signal optical system. More importantly, maintaining and achieving beam aiming, acquisition, and tracking functions at the micro-soliton level becomes an immense challenge. The downlink pertains to the connection between the laser communication transmitter, positioned on the satellite in the upper atmosphere, and the ground receiver at the base station. During free space transmission, the light beam maintains its quality due to minimal deflection. However, upon entering the atmosphere, it encounters attenuation, turbulence, and deflection effects. Notably, the atmospheric channel's impact on the downlink is lesser compared to the uplink, making ground reception relatively easier.

Change4: Page 14, Line 440-453

  1. Song Jiaxue , Chen Chunyi, Yao Haifeng, et al. Study on the coupling efficiency probability distribution of single mode fiber disturbed by turbulence [J].Progress in Lasers and Optoelectronics, 2021,58 (19): 1906002.
  2. Ashraf M , Idrus S , Iqbal F ,et al. Disaster-Resilient Optical Network Survivability: A Comprehensive Survey[J].Photonics, 2018, 5(4):35
  3. Maher M M , El-Samie F E A , Zahran O .A Novel Dynamic Bandwidth Allocation for the Integrated EPON and WiMAX Based on Auction Process[C].2019.
  4. Morio, Toyoshima, Hideki, et al. Atmospheric turbulence-induced fading channel model for space-to-ground laser communications links[J].Optics Express, 2011.
  5. Perlot N , Horwath J , Juengling R .Modelling Wind in Simulations of Atmospheric Optical Propagation[C].Conference on Free-Space Laser Communication Technologies XVII; 20050125-26;
  6. Lin Li , Ning Ji , et al. Design and Experimental Demonstration of an Atmospheric Turbulence Simulation System for Free-Space Optical Communication[J]. Photonics2024, 11(4),334.
  7. Chen J , Huang Y , Cai R ,et al. Free-Space Communication Turbulence Compensation by Optical Phase Conjugation[J].IEEE Photonics Journal, 2020, 12(5):1-11.

1.4. Change5: Page 5, Line 174-179

Figure. 1 Schematic diagram illustrating the coupling of a space laser to a single-mode fiber in a satellite-to-ground laser communication downlink.

1.5. Change6: Page 5, Line 183:Page6, Line 207

(1)

(7)

1.6. Change7: Page 8, Line 250-256

The Hufnagel-Valley (H-V) model is a widely-used atmospheric turbulence propagation model. It describes propagation attenuation and phase disturbances in the atmosphere, considering the effects of varying atmospheric conditions like wind speed, temperature, and humidity on turbulence propagation. This model offers a means to quantify and predict the impact of atmospheric turbulence on communication links. For slant transmission, the atmospheric refractive index structure constant in the Hufnagel-Valley model is expressed as follows [3,9]:

1.7. Change8: Page 9, Line 300-301,Page 10, Line 311-312

(c)Power spectrum of received signal

1.8. Change9: Page 13, Line 367-383

In this paper, we simulate the transmission process of optical signals in satellite-ground laser communication links through numerical calculations and simulation experiments, focusing on coupling efficiency. Additionally, we analyze how changes in wind speed affect the phase of optical signals. Our findings reveal that atmospheric wind speed alters the amplitude and phase characteristics of laser beam propagation within atmospheric turbulence. Increasing wind speed often intensifies atmospheric turbulence, which leads to changes in the refractive index. These changes, in turn, alter the phase and amplitude of the light beam, causing it to deviate from its predetermined path. This deviation disrupts the directional transmission of the optical signal, alters the shape of the power spectrum's probability density model, reduces the time autocorrelation coefficient of the optical signal, and ultimately decreases coupling efficiency. Thereby, the reliability of the communication link is affected. In this paper, the effectiveness of the proposed method in satellite-ground laser communication is verified through fitting analysis, providing strong support for turbulence coupling efficiency studies. However, this method still faces limitations regarding real-time application, stability, and performance verification. In the future, we should delve deeper into the mechanisms of atmospheric turbulence's influence. Additionally, we need to develop cutting-edge optical technology, beam control methods, and explore multi-beam parallel transmission to propel the continued advancement of laser communication technology.

1.9. Change10: Page 13, Line 380-383

This method still faces limitations regarding real-time application, stability, and performance verification. In the future, we should delve deeper into the mechanisms of atmospheric turbulence's influence. Additionally, we need to develop cutting-edge optical technology, beam control methods, and explore multi-beam parallel transmission to propel the continued advancement of laser communication technology.

Round 2

Reviewer 1 Report

Comments and Suggestions for Authors

The responses to the comments are not sufficient. More precisely:

- Absence of confirmation from experimental data suggests potential flaws in the modeling. 

- The selection of the H-V model for atmospheric turbulence remains unclear. Why is it prefered against other turbulence models? 

- Additionally, the formatting requires refinement. There is inconsistency even in the presentation of measurement's units (alternation between plain text and italics).

Comments on the Quality of English Language

N/A

Author Response

April 16, 2024

Editorial Department of photonics

Dear Reviewers,

Thank you for giving us an opportunity to revise our manuscript, we appreciate editor and reviewers very much for their positive and constructive comments and suggestions on our manuscript entitled “Simulation Method for The Impact of Atmospheric Wind Speed on Optical Signals in Satellite-Ground Laser Communication Link” ( ID: photonics-2946911). We have carefully reviewed the comments provided by reviewers and have incorporated these valuable suggestions into comprehensive revisions of the paper.

(Note: Modifications to the identical sections in the manuscript, highlighted in red font, address same concerns raised by reviewers

Response:

1.

It's crucial to acknowledge that models serve as simplified representations of complex real-world phenomena. They are often based on assumptions and simplifications that may not perfectly capture all aspects of the system being studied. Therefore, the absence of confirmation from experimental data should be seen as an opportunity for further investigation, improvement, and potentially the development of more accurate models that align with empirical observations.

  1. Change1: Page8, Line 250-268

The selection of the H-V (Hufnagel-Valley) model for atmospheric turbulence is preferred in Satellite-ground laser communication applications due to its simplicity and versatility. Here are a few reasons why it is often chosen over other turbulence models:1) Empirical validity: The H-V model has been extensively validated through experimental data and observations in a wide range of atmospheric conditions. It provides a good approximation of turbulence effects in many practical scenarios, such as Satellite-ground laser communication and optical systems.2) Simplified representation: Compared to more complex turbulence models like the Kolmogorov model or von Kármán model, the H-V model offers a simplified representation of atmospheric turbulence. It assumes a single-layered, isotropic, and homogeneous structure, making it easier to implement in simulations and analytical calculations.3) Computational efficiency: The simplicity of the H-V model leads to computational efficiency, making it suitable for real-time simulations and system-level analysis. Other turbulence models that consider finer details or specific atmospheric characteristics may require more computational resources and time.4) Broad applicability: The H-V model's versatility allows it to be applied in various fields, including wireless communication, radar systems, and astronomy. It provides a reasonable estimate of turbulence-induced effects on signal propagation, such as scintillation, fading, and beam spreading. For slant transmission, the atmospheric refractive index structure constant in the Hufnagel-Valley model is expressed as follows [3,9]:

  1. Change2: Page9, Line 299-300; Page11, Line356-357

Table 1. The system model parameters.

L0

l0

L

   h    

  ?

   D

  f

    ?

  ω

Wm

1.7×10-14m-2/3

1m

0.1m

300000m

2000m

45℃/180

0.25m

0.6m

1.55×10-6m

0.1m

5.25×10-6m

Table 2. The data corresponding to the mean and standard deviation used for fitting the received signal optical power probability density under different wind speed conditions

w

mean

std

v=1m/s

v=2m/s

v=3m/s

v=1m/s

v=2m/s

v=3m/s

0.6035

21m/s

0.001225

0.001925

0.001575

0.0006134

0.001018

0.0008443

0.460

30m/s

0.0028

0.00211

0.00275

0.001544

0.001198

0.001515

Reviewer 2 Report

Comments and Suggestions for Authors

The revised manuscript is much better than before, which could be accepted in present form.

Author Response

Thank you for giving us an opportunity to revise our manuscript, we appreciate editor and reviewers very much for their positive and constructive comments and suggestions on our manuscript entitled “Simulation Method for The Impact of Atmospheric Wind Speed on Optical Signals in Satellite-Ground Laser Communication Link” ( ID: photonics-2946911)

Reviewer 3 Report

Comments and Suggestions for Authors

Accepted

Comments on the Quality of English Language

Minor

Author Response

(The authors gave the same response as above.)

Round 3

Reviewer 1 Report

Comments and Suggestions for Authors

Ν/Α

Comments on the Quality of English Language

Ν/Α

Author Response

Dear Reviewer,

 We would like to express our sincere gratitude for your review of our manuscript and valuable suggestions. We truly appreciate the time and effort you have dedicated to this review process.

Based on your advice and guidance, we have carefully considered and revised the manuscript. We have incorporated your suggestions and made appropriate modifications to address the issues raised. The revised manuscript is now more accurate, clear, and rigorous.

We would also like to extend our sincere apologies. It has come to our attention that there were inaccuracies or misleading statements in our previous version that went unnoticed. We deeply regret this oversight and appreciate your guidance and promptings that helped us correct these mistakes.

Your expertise and experience have greatly influenced our research work, and we are grateful for the help and guidance you have provided. We will carefully consider the additional suggestions you mentioned in the review and give them full consideration in the next round of revisions. We believe that with your guidance, our manuscript will further improve and be enhanced.

Once again, we sincerely thank you for your reviewing work. We look forward to continuing our collaboration and hope that you will be satisfied with the revised manuscript we eventually submit.

(Note: Modifications to the identical sections in the manuscript, highlighted in red font, address same concerns raised by reviewers.)

Reviewer #1:

  1. Absence of confirmation from experimental data suggests potential flaws in the modeling. 
  2. The selection of the H-V model for atmospheric turbulence remains unclear. Why is it preferred against other turbulence models? 

Respond to the Reviewers’ comments.

The revised paragraphs based on REVIEWER #1 are labeled in red-colored font.

Response:

  1. Change1: Page9, Line 298-303; Page12, Line 359-363; Page13, Line 401-406

Because only a limited number of samples can be used in the simulation experiment, there may be some errors between the correlation of the random sample signal and the theoretical correlation. In order to solve this problem, 200 simulation experiments were repeated, and the normalized autocorrelation function coefficients were calculated according to the results.

It is observed from Fig. 4 and Fig. 5 that the optical signal temporal autocorrelation coefficient model and the received optical signal power spectrum probability density model simulated by the Hufnagel-Valley (H-v) model are basically consistent with the fitted model, indicating that the Hufnagel-Valley (H-v) model can accurately represent the turbulence effect at different wind speed.

However, this method still faces limitations regarding real-time application, stability, and performance verification. In the future, we should delve deeper into the mechanisms of atmospheric turbulence's influence. Additionally, we need to develop cutting-edge optical technology, beam control methods, and explore multi-beam parallel transmission to propel the continued advancement of laser communication technology.

  1. Change2: Page8, Line 250-268

The selection of the Hufnagel-Valley (H-V) model for atmospheric turbulence in satellite-ground laser communication applications is preferred due to its simplicity and versatility. There are several reasons why it is often chosen over other turbulence models:1) Empirical validity: The H-V model has been extensively validated using experimental data and observations in various atmospheric conditions. It provides a good approximation of turbulence effects in practical scenarios, such as satellite-ground laser communication and optical systems.2) Simplified representation: Compared to more complex turbulence models like the Kolmogorov or von Kármán models, the H-V model offers a simplified representation of atmospheric turbulence. It assumes a single-layered, isotropic, and homogeneous structure, making it easier to implement in simulations and analytical calculations.3) Computational efficiency: The simplicity of the H-V model leads to computational efficiency, making it suitable for real-time simulations and system-level analysis. Other turbulence models that consider finer details or specific atmospheric characteristics may require more computational resources and time.4) Broad applicability: The versatility of the H-V model allows it to be applied in various fields, including wireless communication, radar systems, and astronomy. It provides a reasonable estimate of turbulence-induced effects on signal propagation, such as scintillation, fading, and beam spreading.

For slant transmission, the atmospheric refractive index structure constant in the Hufnagel-Valley model is expressed as follows:[3,9]: